# Cryo-EM structure of amyloid fibrils formed by the entire low complexity domain of TDP-43

Qiuye Li [1], W. Michael Babinchak [1] & Witold K. Surewicz [1 ✉]

Amyotrophic lateral sclerosis and several other neurodegenerative diseases are associated with brain deposits of amyloid-like aggregates formed by the C-terminal fragments of TDP-43 that contain the low complexity domain of the protein. Here, we report the cryo-EM structure of amyloid formed from the entire TDP-43 low complexity domain in vitro at pH 4. This structure reveals single protofilament fibrils containing a large (139-residue), tightly packed core. While the C-terminal part of this core region is largely planar and characterized by a small proportion of hydrophobic amino acids, the N-terminal region contains numerous hydrophobic residues and has a non-planar backbone conformation, resulting in rugged surfaces of fibril ends. The structural features found in these fibrils differ from those previously found for fibrils generated from short protein fragments. The present atomic model for TDP-43 LCD fibrils provides insight into potential structural perturbations caused by phosphorylation and disease-related mutations.

[1] Department of Physiology and Biophysics, Case Western Reserve University, Cleveland, OH, USA. ✉email: wks3@case.edu

Proteinaceous brain inclusions containing the transactive response DNA-binding protein of 43 kDa (TDP-43) are a pathological hallmark of amyotrophic lateral sclerosis (ALS) and frontotemporal lobar degeneration (FTLD)[1–3]. Similar inclusions have also been found in several other neurodegenerative disorders, including Alzheimer's disease, cerebral age-related TDP-43 with sclerosis, dementia with Lewy bodies, hippocampal sclerosis, Huntington's disease, and chronic traumatic encephalopathy, among others[2,3].

TDP-43 consists of an N-terminal domain, two RNA recognition motifs, and a C-terminal low-complexity domain (LCD) that maps to residues ~267–414 and is rich in Gln/Asn and Gly residues[3]. Pathological inclusions in TDP-43 proteinopathies typically consist of C-terminal fragments of different sizes, the main component of which is the LCD[4–6]. The latter domain, which contains most disease-associated mutation sites[2,3], is believed to drive the aggregation process[7,8]. Even though morphologically distinct TDP-43 aggregates have been observed in patient-derived histopathological samples[1,9], many of them appear to have fibrillar structure and stain with amyloid-specific dyes[10–12]. Furthermore, recent reports indicate that these aggregates can self-propagate by the prion-like mechanism[13–16]. Consistent with these findings, polypeptides corresponding to TDP-43 LCD or its fragments have been shown to form amyloid fibrils in vitro[3,8,17–19].

In contrast to recent progress in high-resolution structural studies of other amyloids[19–36], however, similar studies with TDP-43 are limited to fibrils formed by relatively short fragments of the protein[19]. To bridge this critical gap, here we report a near-atomic-resolution cryo-electron microscopic (cryo-EM) structure of fibrils formed in vitro by the entire LCD domain of TDP-43. Using our structural model for wild-type protein, we also assess the potential impact of disease-associated mutations and phosphorylation of individual Ser residues.

## Results

**Fibril morphology as assessed by atomic force microscopy (AFM).** To optimize sample preparation for cryo-EM studies, TDP-43 LCD fibrils were first analyzed by AFM. Consistent with the previous report[19], fibrils formed at pH 7 were found to clump together, precluding detailed morphological analysis. This clumping tendency was reduced for fibrils prepared at pH 6, even though they were still not sufficiently dispersed to allow high-resolution cryo-EM studies. When analyzed by AFM, the later fibrils displayed two different morphologies, one twisted and one without a twist (Supplementary Fig. 1a). The twisted fibrils were left-handed, showing in AFM images a maximum height of 7.2 ± 0.3 nm and periodicity of 49.0 ± 6.0 nm.

Fibrils formed under mildly acidic conditions (pH 4) were much better dispersed and more suitable for cryo-EM studies. Akin to pH 6 fibrils, these fibrils displayed two different morphologies, one twisted and one lacking such a twist (Supplementary Fig. 1b). In an attempt to select one preferential morphology, four rounds of sequential seeding reactions were performed by adding preformed, sonicated fibrils (10% w/w) to freshly prepared, non-aggregated protein under the same buffer conditions. Despite these efforts, akin to the first-round fibrils, the final sample used for cryo-EM studies also contained two distinct fibril morphologies (Supplementary Fig. 1c). The relative populations of the two fibril types did not change significantly during sequential seeding reactions, with the twisted species in the non-seeded reaction and last-round seeded reaction accounting for 51 ± 16 and 45 ± 10% of the entire fibril population, respectively. The morphology of pH 4 twisted fibrils was very similar to that of pH 6 counterparts: they were left-handed and

characterized by the height maximum of 7.8 ± 0.9 nm and periodicity of 46.3 ± 2.8 nm.

**Cryo-EM structure of TDP-43 LCD fibrils.** Consistent with AFM data, cryo-EM images of fibrils formed at pH 4 revealed two types of morphologies, one of them showing a helical twist and the second one lacking such a twist (Supplementary Fig. 2a, b). Helical reconstruction of twisted fibrils (all of which share the same fold) allowed us to determine a density map with a nominal resolution of 3.2 Å (Supplementary Table 1 and Supplementary Fig. 2c). Each twisted fibril consists of a single left-handed protofilament in which subunits stack along the fibril axis with a helical rise of 4.73 Å and a helical twist of −1.66° (Fig. 1a). A near-atomic-resolution structural model could be unambiguously built for the fibril core, which maps to residues 276–414 of TDP-43 (Fig. 1b and Supplementary Table 1).

This core consists of 14 β-strands linked by relatively rigid turns and loops (Fig. 1c, d). Each protein subunit can be divided into two regions that differ with regard to an overall amino acid composition as well as the backbone geometry (Figs. 1d and 2a, b). The C-terminal region (residues 344–414, strands β9–β14) is characterized by a small proportion of hydrophobic amino acids. Relatively long β-strands pack mostly via polar interfaces and many residues within strands β9, β10, β11, and β13 are engaged in steric–zipper interactions (Figs. 1d and 2a). The stacking of subunits within this region is largely maintained by a network of intermolecular backbone H-bonds between β-strands, with additional stabilization through H-bonds between numerous stacked Gln and Asn side chains (Supplementary Fig. 3). This C-terminal region is largely planar, with most β-strands packed through interactions within the same subunit (Fig. 2b).

By contrast, the N-terminal region of LCD (residues 276–343, strands β1–β8) is rich in hydrophobic amino acids and contains mostly short β-strands that are not involved in steric–zipper interactions. Instead, strands and turns pack against each other in all directions, allowing non-polar side chains to be tightly packed and buried, with a hydrophobic interface between the 311–318 and 336–341 segments (Fig. 1d, e). Side chains on the opposite side of the 336–341 segment form another hydrophobic interface with residues 381–383 within the LCD C-terminal region. Altogether, these three segments make up a large hydrophobic core. A second, smaller hydrophobic core involves residues within the Ala- and Met-rich 321–330 segment that pack against Phe283, Phe289, and Trp334 (Figs. 1e and 2a). To facilitate such a tight packing of hydrophobic residues, the backbone in this region of each subunit makes numerous turns within the $x$–$y$ plane while also extending along the $z$-axis over a distance of ~22.4 Å (Fig. 2b). As a result, each subunit (i) not only interacts with the layer directly above (i + 1) and below (i − 1) but also with layers up to (i + 3) and (i − 3), as illustrated in Fig. 2c for the 311–327 segment. Furthermore, some of the interactions are between stacks of hydrophobic residues, which are arranged in a staggered fashion (Fig. 2d, e). Thus, even though only ~40% of residues are involved in intermolecular H-bonds within the cross-β motif, fibrils are likely further stabilized through the interlayer interactions between side chains.

The non-planar backbone conformation of TDP-43 LCD subunits (which is not unusual among amyloids[22,28,35]) results in rugged surfaces of fibril ends with well-defined "ridges" and "grooves". The ridges are composed mostly of water-exposed hydrophobic residues, with one end containing 15 such residues from the top three subunits [M311 (m, m − 1, m − 2), F313 (m, m − 1, m − 2), F316 (m, m − 1), I318 (m, m − 1), M322 (m), A324 (m), M337 (m), M339 (m), and M405 (m)], and the other end containing nine hydrophobic residues from the bottom three

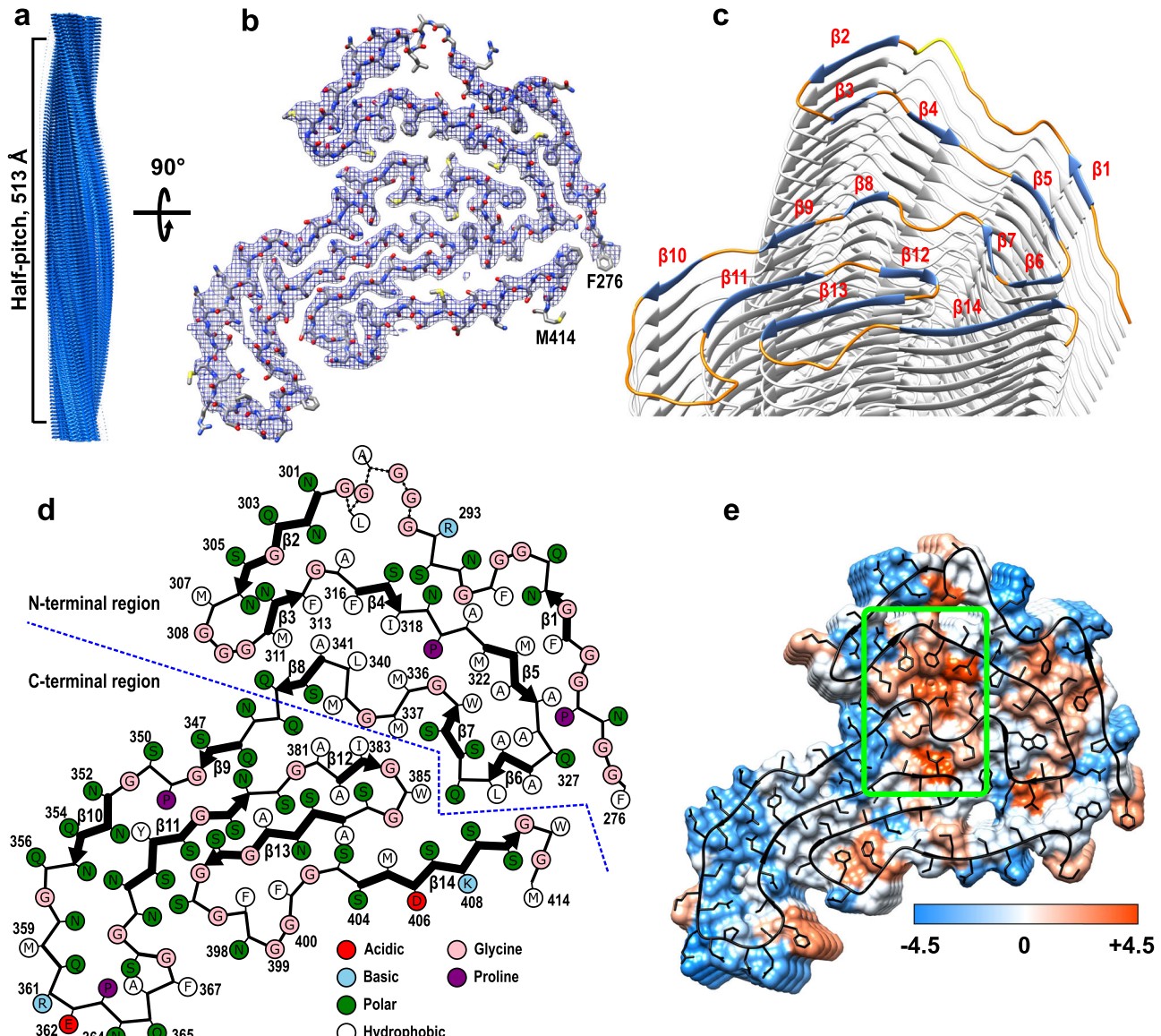

**Fig. 1 Cryo-EM structure of TDP-43 LCD fibrils. a** Cryo-EM density map showing a left-handed helix with a half-pitch of 513 Å. Repeating densities representing β-strands along the z-axis with a 4.73-Å interval indicate a parallel in-register β-sheet architecture. **b** The atomic model and superimposed cryo-EM map of one cross-sectional layer of each fibril (top view). **c** Tilted cross-section of a TDP-43 LCD fibril. β-Strands in the top subunit are shown in blue, highly ordered turns in orange, and the less ordered region in yellow. **d** Schematic representation of one cross-sectional layer of the amyloid core, with β-strands shown as thicker arrows and the less ordered region (residues 295–299) marked as dotted lines. **e** Hydrophobicity of the fibril cross-section, with hydrophobicity levels colored according to Kyte–Doolittle[54]. A major hydrophobic core (green box) is made up of the 311–318, 336–341, and 381–383 segments.

subunits [A341 (n, n + 1, n + 2), L340 (n, n + 1), M336 (n), A381 (n), I383 (n), and A388 (n)] (Fig. 3).

## Discussion

Rapidly accumulating cryo-EM data for different types of amyloid fibrils point to a large structural polymorphism of fibrils formed even by the same protein. In the present study, we determined a near-atomic-resolution structure of one polymorphic form of fibrils formed by the entire LCD of TDP-43, a protein associated with many neurodegenerative diseases. The second type of fibrils observed in our samples lacks any twist, precluding high-resolution structural characterization by cryo-EM. Even though the present structure has been determined for fibrils formed under mildly acidic conditions (pH 4), similar morphology of

fibrils generated at pH 6 suggests that the latter fibrils may be structurally similarity also at the atomic level.

A notable feature of the present structure of TDP-43 LCD fibrils is the size of the amyloid core that is comprised of 139 residues (out of 148 present in the LCD). To the best of our knowledge, the size of this core region is the largest among all amyloid fibrils structurally characterized to date[37]. Interestingly, in contrast to most other reported fibril structures that contain two or more protofilaments[37], TDP-43 LCD fibrils consist of a single protofilament. This is, however, not an unprecedented feature, as single protofilament structures have been reported for fibrils formed by antibody light chains[32,33], transthyretin[34], FUS LCD[20], and some polymorphic forms of tau[24–26] and α-synuclein[29].

Another notable feature of TDP-43 LCD fibrils is a highly non-planar backbone conformation within the N-terminal

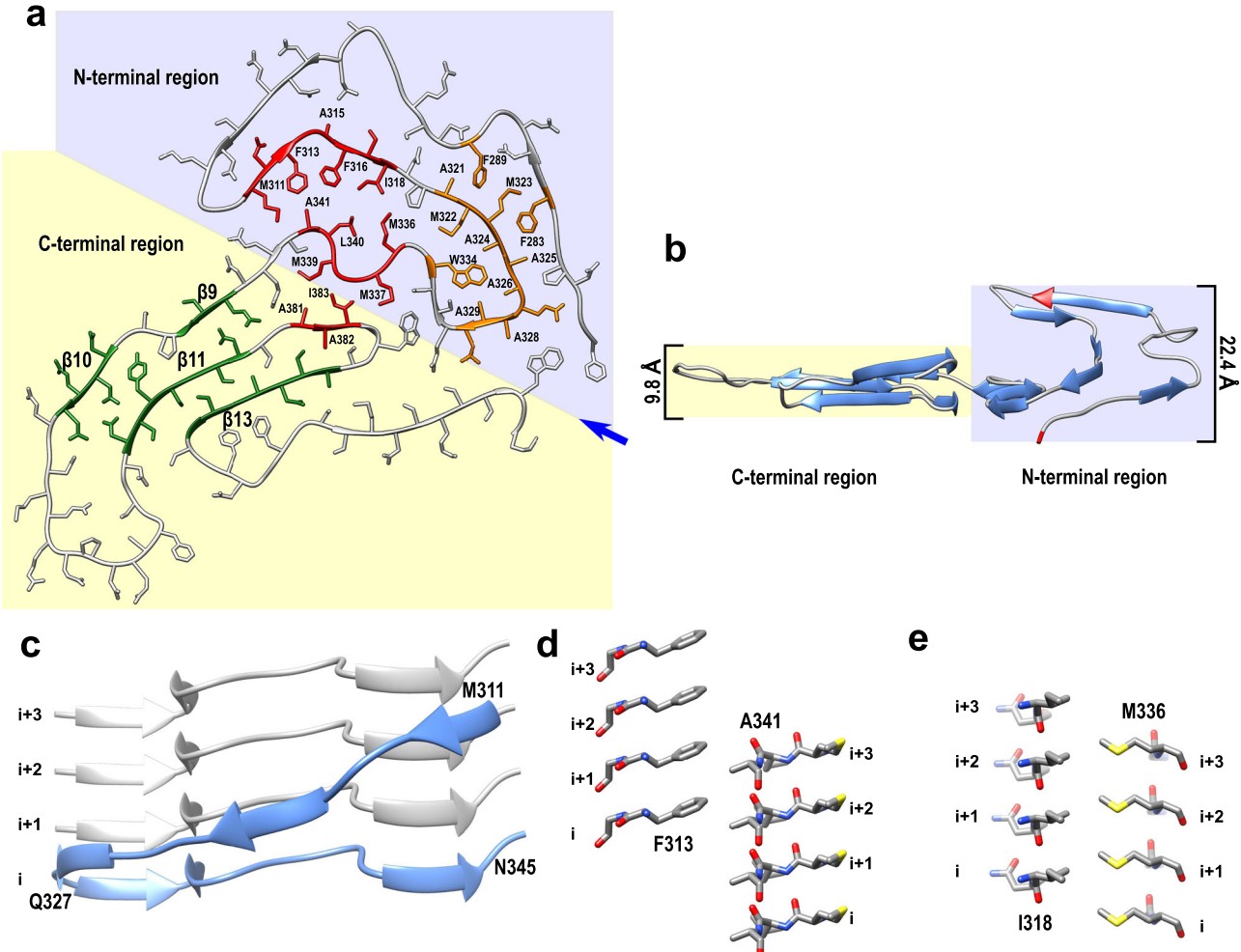

**Fig. 2 Differences between the N- and C-terminal regions of the amyloid core of TDP-43 LCD fibrils. a** Top view of one subunit within the fibril. Large (red) and smaller (orange) hydrophobic cores are present within the N-terminal part, stabilizing this region. The C-terminal region is stabilized largely by steric–zipper interactions involving side chains within strands β9, β10, β11, and β13 (green). **b** Side view of one subunit within the fibril (with the view angle indicated by the blue arrow in **a**). In contrast to a largely planar C-terminal region, the N-terminal region extends along the fibril axis over the distance of 22.4 Å. The lowest (Phe276) and the highest (Asn306) points are marked in red. **c** Side view of the 311–327 segment extending along the fibril axis. This segment in subunit i interacts with the 327–345 segment in subunits i + 1, i + 2, and i + 3. **d**, **e** Side view of interlayer interactions between F313 and A341 (**d**) and I318 and M336 (**e**). Layers of side chains are packed in a staggered fashion, resulting in a very compact hydrophobic core.

region of the amyloid core. This results in rugged surfaces of fibril ends with many water-exposed hydrophobic residues. Such highly hydrophobic, rugged surfaces of fibril ends may be especially conducive to the recruitment of TDP-43 monomers and their templated conversion into the amyloid conformation. Distinct surfaces at opposite ends could also result in fibril polarity, with different elongation rates at each end. Furthermore, detailed structural characterization of these surfaces (Fig. 3) may provide a starting point for a rational design of TDP-43 amyloid inhibitors.

A recent study reported cryo-EM structures of fibrils formed by two relatively short fragments of TDP-43 LCD[19]. The first fragment (residues 311–360) formed polymorphic amyloid structures with a common motif described as a dagger fold, in which residues from Phe313 to Ala341 form tight hydrophobic interaction. A sharp (~160°) kink at Gln327 defines the tip of the dagger (Supplementary Fig. 4a). The fold adopted by this segment in our structure of fibrils formed by the entire LCD is substantially different, with no sharp kink at Gln 327. Instead, there is a ~90° turn at this residue, followed by another ~90° turn at Gln331, such that the overall fold shows little resemblance to the

dagger motif (Supplementary Fig. 4b). Fibrils formed by the second fragment (residues 286–331 with A315E mutation) consisted of four protofilaments that each contain another common motif characterized as R-shaped fold (Supplementary Fig. 4c). It was proposed that a similar fold (stabilized by hydrophobic interactions between Ala315, Ala297, and Phe313) would be adopted by this segment without the mutation[19]. Again, the fold within this region of fibrils formed by the entire LCD is quite different: Ala297, Phe313, and Ala315 are not in close contact, and the local conformation is dictated by interactions with other parts of the molecule (Supplementary Fig. 4d).

Over 30 point mutations within the TDP-43 LCD are associated with ALS and FTLD[2,3]. The mechanisms by which these mutations facilitate disease are poorly understood. Mapping the pathogenic mutations on the structure of wild-type TDP-43 LCD fibrils revealed that ~50% of them are not compatible with this structure due to severe steric clashes within tightly packed segments of the protein, introduction of charges into the dehydrated fibril interior, or both (Fig. 4a). Thus, these mutations will likely result in substantially different fibril structures, and this may affect the disease phenotype.

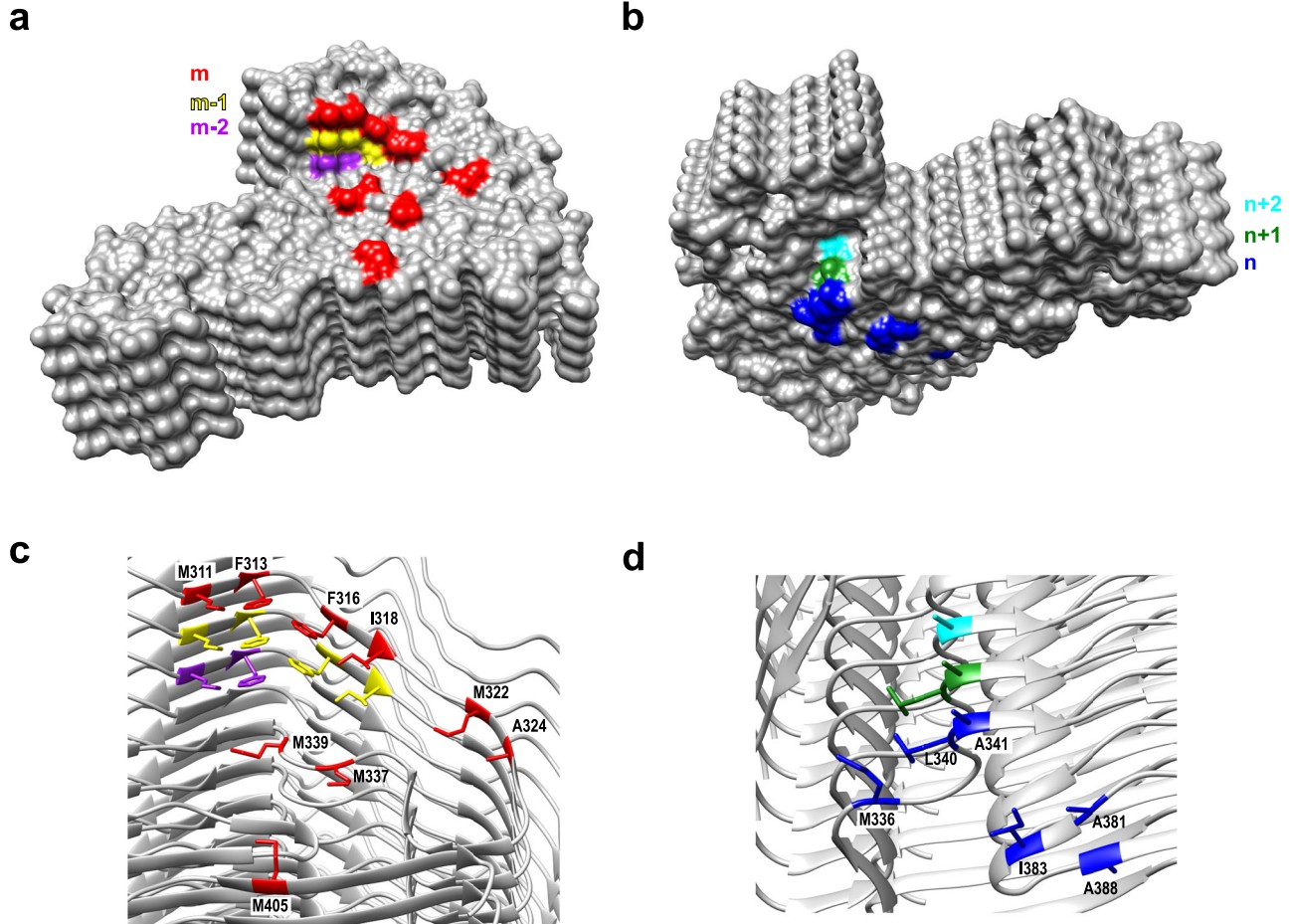

**Fig. 3 Rugged surface of the top and bottom ends of TDP-43 LCD fibril. a, b** Surface representation of fibril ends with solvent-accessible hydrophobic residues from different layers marked in different colors. **c, d** Ribbon representation of the structure at both fibril ends. Exposed hydrophobic residues from different layers are labeled using the same color scheme as in **a, b**. Fifteen hydrophobic side chains from three different layers [M311 (m, m − 1, m − 2), F313 (m, m − 1, m − 2), F316 (m, m − 1), I318 (m, m − 1), M322 (m), A324 (m), M337 (m), M339 (m), and M405 (m)] are exposed to water at the top end. Nine hydrophobic side chains [A341 (n, n + 1, n + 2), L340 (n, n + 1), M336 (n), A381 (n), I383 (n), and A388 (n)] are exposed at the bottom end.

One of the features of ALS and FTLD pathology is phosphorylation of Ser residues within the C-terminal part of the TDP-43 LCD, with the consensus pathological phosphorylation sites at Ser403, Ser404, and Ser409/410, and additional disease-specific phosphorylation sites at Ser379 and Ser369[3,38]. The full phosphorylation landscape and population of molecules phosphorylated at specific sites are, however, unknown. Furthermore, non-phosphorylated protein seems to be also present in pathological inclusions, as indicated by staining with antibodies raised against non-phosphorylated C-terminal epitopes[39]. Interestingly, while two of the common phosphorylation sites (Ser404 and Ser410) are exposed on the fibril surface, others (Ser403, Ser409, Ser379, and Ser369) are buried in the interior of the fibril structure that we determined for the non-phosphorylated protein (Fig. 4b). Thus, phosphorylation of the latter residues will likely affect the structure of fibrillar aggregates, potentially resulting in a large, phosphorylation site-dependent structural polymorphism. This, in turn, could further influence the disease phenotype. The present structure for non-phosphorylated TDP-43 LCD fibrils provides a necessary foundation for future high-resolution structural studies with fibrils containing protein variants with different phosphorylation patterns.

## Methods

### Protein expression and purification.
The plasmid for bacterial expression of TDP-43 LCD with an N-terminal His-tag was described previously[8]. The protein was expressed overnight using Rosetta™ (DE3) pLysS competent cells (MilliporeSigma) after induction with 1 mM isopropyl β-D-1-thiogalactopyranoside. Cells were collected by centrifugation, lysed by sonication in Buffer A (20 mM Tris-HCl buffer, pH 8, containing 8 M urea, 500 mM NaCl, and 25 mM imidazole), and purified over Ni-charged nitrilotriacetic acid column using 4–5 column volume washes with Buffer A followed by elution with Buffer B (20 mM Tris-HCl buffer, pH 8, containing 8 M urea, 200 mM NaCl, and 250 mM imidazole). Protein was concentrated and purified by high-performance liquid chromatography using a $C_4$ column with acetonitrile gradient in water containing 0.05% trifluoroacetic acid. Protein purity (>95%) was confirmed by gel electrophoresis. Pure protein was flash frozen and lyophilized for later use.

### Fibril formation and morphology analysis using AFM.
Lyophilized protein was dissolved in Milli-Q purified $H_2O$ and passed through an Amicon Ultra centrifugal filter with 100 kDa molecular weight cut-off to remove preformed aggregates[8]. Fibrils were formed at a protein concentration of 30 μM in 20 mM MES buffer, pH 6.0, or 20 mM sodium acetate buffer, pH 4.0. The samples were placed on rotation (~6–8 rpm) at 37 °C. For AFM imaging, 2 μl of fibril suspension was diluted 10-fold in Milli-Q $H_2O$, deposited on freshly cleaved mica substrate and incubated for 5 min. The surface was then washed four times with Milli-Q $H_2O$ and dried under $N_2$. The images were obtained by NanoScope 9.1 using scan assist mode and a silicon probe (spring constant, 40 newtons/m) on a Bruker multimode atomic force microscope equipped with Nanoscope V controller. Image analysis was performed using Nanoscope Analysis 1.5. Fibrils generated from both reactions displayed two different morphologies on AFM images, one twisted and one lacking a twist. The reported height maxima and periodicity of twisted fibrils are based on measurements using 20 randomly selected fibrils in each group. In an attempt to select one preferential morphology in pH 4 fibrils, 4 rounds of sequential seeding reactions were then performed by adding preformed, sonicated pH 4 fibrils (10% w/w) to freshly prepared, non-aggregated protein under the same buffer conditions. Despite these efforts, akin to the first-round fibrils, the final sample used for cryo-

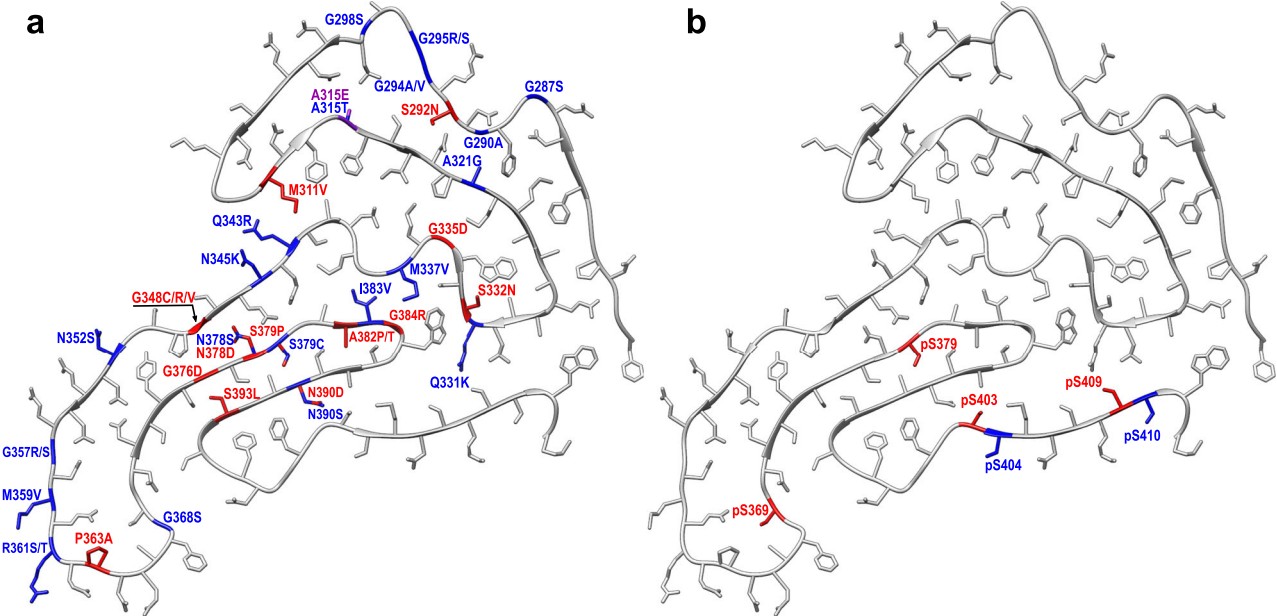

**Fig. 4 Disease-related point mutations and phosphorylation sites mapped on the structure of one subunit of fibrils formed from the wild-type, non-phosphorylated TDP-43 LCD. a** Mutations that are compatible with this structure are labeled in blue. The remaining mutations (labeled in red) are not compatible with the structure determined for wild-type protein fibrils due to steric clashes within tightly packed segments of the protein (S292N, M311V, S332N, G348C/V, S379P, A382P/T, S393L), introduction of charges into the dehydrated fibril interior (N378D, N390D), or both (G335D, G348R, G376D, G384R). P363 mutation would likely interfere with the formation of a turn observed in the fibril structure for the wild-type protein. Given that the 295–299 segment is relatively flexible in the present structure, the compatibility of A315E substitution (labeled in purple) with this structure is difficult to assess. **b** Phosphorylation sites exposed on the surface and those buried inside the structure determined for wild-type TDP-43 LCD fibrils are labeled in blue and red, respectively. Phosphorylation of exposed S410 would require only very small structural rearrangement of the backbone of C-terminal residues.

EM studies also contained two types of fibril morphologies (Supplementary Fig. 1). To assess the percentage of twisted fibrils, the total length of twisted and non-twisted fibrils in three randomly selected 2 µm × 2 µm AFM images was measured.

**Cryo-electron microscopy.** Two hundred mesh lacey carbon grids (Ted Pella) were first coated with 0.1 mg/ml graphene oxide and then with 0.1% poly-lysine as described previously[40,41]. Three microliters of TDP-43 LCD fibril suspension (30 µM) was applied to the coated grid, blotted for 6 s, and plunge-frozen in liquid ethane using a Vitrobot Mark IV (ThermoFisher Scientific). Movies were collected on a Titan Krios G3i microscope (ThermoFisher Scientific) equipped with a Bio-Quantum K3 camera (Gatan, Inc.), with 0.414 Å/pixel at super resolution mode, 42 e−/Å² total dose, and 60 total frames. A total of 6589 micrographs were automatically collected using SerialEM[42] with 6 shots per position. Beam image shift was applied, and defocus range was between −0.8 and −1.5 µm.

**Data processing.** Movies were corrected for drifting and binned by a factor of 2 using MotionCor2[43]. Contrast Transfer Functions (CTF) were estimated by Gctf[44]. All further processing was carried out using RELION 3.1[45–47]. Fibrils with apparent helical twists were manually picked and a total of 294,168 segments were extracted using an overlap of 97% between neighboring segments and a box size of 512 pixels. Segments were first subjected to several rounds of reference-free two-dimensional (2D) classification using $T = 8$ and $K = 100$ to remove poorly defined classes, resulting in 65,075 segments contributing to clear 2D averages. These segments were then used for subsequent three-dimensional (3D) classification employing an initial model of a featureless cylinder generated by relion_helix_toolbox[46]. The initial helical rise (4.73 Å) was calculated from the 2D class layer line profile, and the initial helical twist (−1.64°) was calculated from the crossover distance. The handedness of the helix was determined by AFM images. The tilt of all segments were kept at 90° throughout the 3D processing. Two rounds of 3D classification were performed using $K = 3$ and $T = 4$, resulting in 11,026 segments that contributed to a high-resolution reconstruction. Additional rounds of 3D classification were performed on these segments using a single class and increasing $T$ value (4, 8, 20, 40, and 100) with local optimization of helical twist and rise. In the last round of 3D classification, β-strands were well separated and large side chains could be resolved. The model and data were then used for high-resolution gold-standard 3D refinement. Iterative Bayesian polishing[47] and CTF refinement[48] were performed to further improve the resolution. The overall resolution was calculated to be 3.2 Å from Fourier shell correlations at 0.143 between two independently refined half-maps. Refined helical symmetry

(twist = −1.66°, rise = 4.73 Å) was imposed on the post-processed map for further model building.

**Model building.** An initial model was built in Coot[49] using large side chains of ³⁴³QQNQ³⁴⁶ segment as a guide. Five chains were built at the central region of the density map, which covered all types of intermolecular contact within the map. The model was then subjected to iterative real-space refinements in PHENIX[50]. At later stages, segments favoring β-strand conformation were identified and the direction of backbone oxygen and nitrogen atoms were adjusted manually to facilitate hydrogen bonding in β-sheets. Such restraints were also implemented in the subsequent refinements. After real-space refinement, side-chain orientations were manually adjusted to ensure energy-favored geometry. The final model was validated using the comprehensive validation method in PHENIX[50]. A map containing five copies of subunits was extracted manually from the reconstructed map using the UCSF Chimera package[51] to calculate Fourier shell correlations between the map and the atomic model.

The density of the TDP-43 LCD segment 295–299 was weaker than that of the rest of the molecule, possibly due to the less-ordered structure of this segment. In this region, we displayed the map at a low contour level and built the model manually. After automatic refinement in PHENIX[50], this region showed no significant clashes or Ramachandran outliers. It has been reported that such less-defined turns or loops of an amyloid core may adopt more than one conformation[30]. In our study, due to the relatively low number of particles, we were not able to classify and determine different structures in this less-defined region. Thus, the structure of this segment in our refined model may represent the average of multiple conformations.

When assessing structural compatibility of disease-related mutations and phosphorylation at individual Ser residues with our structural model determined for wild-type, non-phosphorylated protein, the Reduce[52] and Probe[53] programs were used to test for severe steric clashes (>1 Å overlap) in the absence of the backbone movement.

**Reporting summary.** Further information on research design is available in the Nature Research Reporting Summary linked to this article.

## Data availability

The cryo-EM map of TDP-43 LCD amyloid fibrils has been deposited to the Electron Microscopy Data Bank (EMDB) under the accession code EMD-23059. The coordinates

of the corresponding model have been deposited to the Protein Data Bank (PDB) under accession code 7KWZ. Other data are available from the corresponding author upon reasonable request.

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

## Acknowledgements

This work was supported by NIH grants RF1 AG061797 and R01 GM094357 and R01 NS103848 and CWRU pilot grant CA-CryoPilot2019-2499971. We thank Kunpeng Li for help with cryo-EM acquisition of cryo-EM data, Gunnar F. Schröder and Xinghong Dai for discussions regarding data processing and model building, respectively, and

Sudha Chakrapani for comments on the manuscript. We are grateful to the Cryo-Electron Microscopy Core at the CWRU School of Medicine (especially Dr. Sudha Chakrapani and Dr. Kunpeng Li) for the access to Cryo-EM instrumentation.

## Author contributions

Q.L. and W.K.S. designed the study. Q.L. collected EM data and performed image processing and model building. W.M.B. purified the protein and prepared fibrils. Q.L. and W.K.S. wrote the manuscript.

## Competing interests

The authors declare no competing interests.
