## [Peer Review File · Nature Communications]

REVIEWER COMMENTS

Reviewer #1 (Remarks to the Author):

The manuscript by Li et al. reports the cryoEM fibril structures of TDP-43 low-complexity domain (LCD). Amyloid aggregation of TDP-43 was commonly observed in different Neurodegenerative diseases including ALS, FTD, PD and AD, and is closely associated with disease progression. Structural investigation of how TDP-43 assembles into pathological amyloid fibrils is of importance towards understanding TDP-43 pathology in disease. The newly reported structure is formed by the entire TDP-43 LC and exhibits totally distinct structures compared to the fibrils formed by different TDP-43 LC segments previously determined (Cao et al. NSMB. 2019). Thus, this work is one step forward to revealing the structural basis of fibrillation of full-length and the pathological 25-35kd CTD of TDP-43. However, in order to strengthen the manuscript, the authors need to address my concerns listed below.

1. The TDP-43 LC fibril used in this study was formed in sodium acetate buffer pH4.0 which is away from physiological condition. Given the extremely high structure polymorphs of fibrils even formed by the same protein (e.g. Tau and Abeta), it could be helpful to prepare fibrils close to neutral pH and briefly compare their fibril morphology with the fibrils formed in pH4.0 using AFM, TEM, or fibril diffraction method to emphasize that the reported pH4.0 fibril structure represents a biological relevant fibril structure of TDP-43 LC.
2. The authors showed two types of fibril morphologies co-existing in the cryoEM sample (Fig.S1). Only the twisted one was determined in this study. The authors might need to provide percentage of two different species in the original fibril sample as well as each from the four rounds of sequential seeding procedure.
3. The large difference between the two fibril ends is quite an interesting and important feature of this TDP-43 LC fibril structure. The authors may need to do more carefully structural analysis and comparison of the two fibril ends. Rather than simply showing a very brief and confusing surface representation of two ends in Fig.S4.
4. To dig in the biological relevance of this fibril structure, the authors mapped disease mutations on the fibril structure and classify those mutations into two distinct groups (Fig. S6a), which is indeed helpful. However, the authors need to carefully check each mutation listed in two groups. A few mutations were put in the wrong group. For instance, A315E should most likely disrupt the fibril core structure. P363A may not form turn observed in the fibril structure.
5. The authors mentioned that the handedness of the helix of fibril structure was determined by AFM, but didn't show any AFM images. The authors need to provide AFM data to support this.
6. The authors claim that "the 138-residue core region of TDP-43 LCD fibrils represents the largest amyloid core reported to date", which is indeed impressive. However, the authors need to provide evidences and statistics to support this core is indeed the largest in different fibril structures determined so far.
7. Fig.3 is not informative and a bit confusing. Depending on which region used for doing structure alignment, the overlapped structure patterns could be largely distinct.
8. FSC curves between model and map should be reported.
9. The authors may need to provide a clear intact crossover images from 2D class average in Fig.S1a.

Reviewer #2 (Remarks to the Author):

The manuscript from Qiuye Li, W. Michael Babinchak and Witold K. Surewicz presents structural cryo-EM study of in vitro fibrils of the entire TDP-43 low complexity domain formed at pH 4. Deposits of TDP-43 aggregates are associated with amyotrophic lateral sclerosis and several other neurodegenerative diseases. There are two types of fibrils formed one twisted one without twist. Only the twisted one was reconstructed. The fibril core contains a 138-residue large tightly packed core with different structural features to previously fibrils from short TDP-43 fragments. ~50% of pathogenic mutations on the structure of wild-type TDP-43 LCD fibrils are not compatible with the presented structure. Additionally, phosphorylation of Ser residues within the C-terminus of TDP-43 LCD are a feature of ALS and mostly not compatible with this structure.

The paper clearly written and the methodology is sound. My main concern is the relevance of the fibril for the disease if there is normally a phosphorylation incompatible with the presented structure, otherwise I have only minor revisions and suggestions.

Comments:

1 Main concern: As the authors wrote, a feature of ALS is phosphorylation and several phosphorylated sites would make it incompatible with the presented structure, why do the authors think this structure is relevant for understanding the disease and not a random in vitro structure at pH 4. Sorry if it sounds harsh, but at the moment it reads like a detailed description of an in vitro fibril structure that is not able to form in vivo under relevant disease conditions and is one of two seemingly very distinct structures (twisted and not twisted) under the same growth condition.

2 It would be good to mention that it is an in vitro structure somewhere early in the main text.

3 How long is the full TDP-43 LCD? Only the length of the fibril core is mentioned.

4 Extended Data Fig. 5.: "Representative" one R to many; Also, it would be helpful to have the rest or at least the neighboring residues of full TDP-43 LCD (b,d) in light grey to help localize the fragment in the whole structure.

Point-by point response to reviewers' comments

We wish to thank both reviewers for their constructive comments and suggestions. We believe that addressing these comments have substantially improved the manuscript.

In addition to the revisions made in response to specific comments of the reviewers, we also made several formatting changes requested by the editor (the original manuscript was submitted as a brief Communication without division into individual sections). We now include separate (and substantially expanded) Introduction and Discussion sections. The changes we made in the revised manuscript are marked in red.

Reviewer #1:

The manuscript by Li et al. reports the cryoEM fibril structures of TDP-43 low-complexity domain (LCD). Amyloid aggregation of TDP-43 was commonly observed in different Neurodegenerative diseases including ALS, FTD, PD and AD, and is closely associated with disease progression. Structural investigation of how TDP-43 assembles into pathological amyloid fibrils is of importance towards understanding TDP-43 pathology in disease. The newly reported structure is formed by the entire TDP-43 LC and exhibits totally distinct structures compared to the fibrils formed by different TDP-43 LC segments previously determined (Cao et al. NSMB. 2019). Thus, this work is one step forward to revealing the structural basis of fibrillation of full-length and the pathological 25-35kd CTD of TDP-43. However, in order to strengthen the manuscript, the authors need to address my concerns listed below.

1. The TDP-43 LC fibril used in this study was formed in sodium acetate buffer pH4.0 which is away from physiological condition. Given the extremely high structure polymorphs of fibrils even formed by the same protein (e.g. Tau and Abeta), it could be helpful to prepare fibrils close to neutral pH and briefly compare their fibril morphology with the fibrils formed in pH4.0 using AFM, TEM, or fibril diffraction method to emphasize that the reported pH4.0 fibril structure represents a biological relevant fibril structure of TDP-43 LC.

The reason that fibrils were formed in pH 4 buffer was that, consistent with the previous finding (Cao et al., NSMB 26, 619-27 (2019)), fibrils formed at neutral pH have a strong tendency for clumping that precludes high-resolution structural studies by cryo-EM. We now address this issue directly at the beginning of the Results section. As requested by the reviewer, we also performed comparative analysis (using AFM) of fibrils formed at pH 4 and 6. Twisted fibrils formed under both conditions are morphologically very similar, both with regard to their height as well as periodicity. We include representative AFM images in a new Extended Data Fig. 1 and describe fibril morphology on p. 4 of the revised manuscript.

2. The authors showed two types of fibril morphologies co-existing in the cryoEM sample (Fig.S1). Only the twisted one was determined in this study. The authors might need to provide percentage of two different species in the original fibril sample as well as each from the four rounds of sequential seeding procedure.

As requested by the reviewer, we have quantitatively assessed the percentages of twisted and non-twisted fibrils in our samples, finding no statistically significant differences in these populations in spontaneously formed (non-seeded) fibril preparations and those formed in the last round of sequential seeding procedure. We now provide these percentages on p. 4 of the revised manuscript.

3. The large difference between the two fibril ends is quite an interesting and important feature of this TDP-43 LC fibril structure. The authors may need to do more carefully structural analysis and comparison of the two fibril ends. Rather than simply showing a very brief and confusing surface representation of two ends in Fig.S4.

As recommended by the reviewer, we have expanded the description of structural properties of fibrils ends, identifying water-exposed hydrophobic residues at both ends (p.6). We have also revised the images shown in original Extended Data Fig. 4, using a color coding to indicate the layers containing individual hydrophobic residues that are exposed at both fibril ends. These revised panels are now shown as part of main Fig. 3.

4. To dig in the biological relevance of this fibril structure, the authors mapped disease mutations on the fibril structure and classify those mutations into two distinct groups (Fig. S6a), which is indeed helpful. However, the authors need to carefully check each mutation listed in two groups. A few mutations were put in the wrong group. For instance, A315E should most likely disrupt the fibril core structure. P363A may not form turn observed in the fibril structure.

We appreciate this comment. As now mentioned in the methods section, our classification is based on the presence or absence of side chain steric clashes and introduction of charges into dry interfaces. The effects at the backbone level (which are likely relatively subtle) are more difficult to assess. We have carefully re-analyzed the potential structural impact of individual mutations and slightly revised the relevant figure (now Fig. 4), correcting classification of the G290A mutation (that was incorrectly labeled in red) and P363A mutation (that was incorrectly labeled in blue). With regard to the A315E mutation, the side chain of this residue faces towards the 295-299 segment that is relatively flexible and poorly defined in our cryo-EM structure. Thus, the structural impact of this mutation is difficult to predict. Regarding the P363A mutation, this substitution would indeed likely interfere with formation of a turn, resulting in a structural rearrangement at the backbone level. We comment on these two mutations at the end of the legend to Fig. 4.

5. The authors mentioned that the handedness of the helix of fibril structure was determined by AFM, but didn't show any AFM images. The authors need to provide AFM data to support this.

We now provide AFM images that document left-handedness of twisted TDP-43 LCD fibrils (Extended Data Fig. 1).

6. The authors claim that "the 138-residue core region of TDP-43 LCD fibrils represents the largest amyloid core reported to date", which is indeed impressive. However, the authors need to provide evidences and statistics to support this core is indeed the largest in different fibril structures determined so far.

One way to document this claim could be to include a table containing information regarding all amyloid fibrils the structure of which has been solved to date. However, we believe that a more practical solution (that does not require expansion of the list of references beyond the typical size) is to provide a reference to the website maintained by the Eisenberg lab. This website provides comprehensive, up-to-date information about all amyloid fibril structures, including the size of the core regions. A reference to this website is now provided in the first paragraph on p. 7. We are not sure what kind of statistical analysis could be performed in this regard.

7. Fig.3 is not informative and a bit confusing. Depending on which region used for doing structure alignment, the overlapped structure patterns could be largely distinct.

We agree with the reviewer that our original Fig. 3 was not very informative. We have deleted this figure. Nevertheless, we feel that some structural comparison between fibrils formed by the entire LCD and its short fragments is warranted. We now show this comparison in Extended Data Fig. 4 (formerly Extended Data Fig. 6) that has been revised to include the remaining parts of the structures (in light gray). See also our response to comment #4 of reviewer 2.

8. FSC curves between model and map should be reported.

As requested by the reviewer, the FSC curve between the map and the atomic model has been included in Extended Data Fig 2c.

9. The authors may need to provide a clear intact crossover images from 2D class average in Fig.S1a.

As requested by the reviewer, we now include this image in Extended Data Fig. 2b.

Reviewer #2:

The manuscript from Qiuye Li, W. Michael Babinchak and Witold K. Surewicz presents structural cryo-EM study of in vitro fibrils of the entire TDP-43 low complexity domain formed at pH 4. Deposits of TDP-43 aggregates are associate with amyotrophic lateral sclerosis and several other neurodegenerative diseases. There are two types of fibrils formed one twisted one without twist. Only the twisted one was reconstructed. The fibril core contains a 138-residue large tightly packed core with different structural features to previously fibrils from short TDP-43 fragments. ~50% of pathogenic mutations on the structure of wild-type TDP-43 LCD fibrils are not compatible with the presented structure. Additionally, phosphorylation of Ser residues within the C-terminus of TDP-43 LCD are a feature of ALS and mostly not compatible with this structure. The paper clearly written and the methodology is sound. My main concern is the relevance of the fibril for the disease if there is normally a phosphorylation incompatible with the presented structure, otherwise I have only minor revisions and suggestions.

1 Main concern: As the authors wrote, a feature of ALS is phosphorylation and several phosphorylated sites would make it incompatible with the presented structure, why do the authors think this structure is relevant for understanding the disease and not a random in vitro structure at pH 4. Sorry if it sounds harsh, but at the moment it reads like a detailed description of an in vitro fibril structure that is not able to form in vivo under relevant disease conditions and is one of two seemingly very distinct structures (twisted and not twisted) under the same growth condition.

This comment sounds indeed a little harsh, setting the bar so high that it would preclude publication of many recently published cryo-EM papers on the structure of fibrils formed in vitro from recombinant proteins lacking post-translational modifications found in pathological brain inclusions [e.g., fibrils formed by short TDP-43 LCD fragments (Guenther et al., 2018, *NSMB*, 25, 463-471; Cao et al., 2019, *NSMB* 26, 619-627), α -synuclein (Li et al., 2018, *Nature Commun* 9, 1-10), prion protein (Glynn et al., 2020, *NSMB* 27, 417-423; Wang et al., 2020, *NSMB* 27, 598-602)], or even those not associated with any disease [e.g., SH3 domain fibrils (Röder et al., 2019, *Nature Commun* 10:1-9)]. Nevertheless, we appreciate this comment as it helped us realize that we haven't adequately addressed the issue of phosphorylation in the original manuscript. While phosphorylation of TDP-43 C-terminal Ser residues is recognized as one of

important features of disease pathology, little is known about the phosphorylation landscape in pathological brain inclusions. As in the case of proteins such as tau (Mair et al., *Anal Chem* 88, 3704-14), there are probably great many subpopulations of TDP-43 molecules with distinct phosphorylation patterns. Importantly, one of them is a non-phosphorylated protein (Tome et al., *Acta Neuropathol Commun* 8, 1-22). Thus, the structure of fibrils formed by the latter protein is likely of direct relevance to disease pathology, even though one cannot exclude the possibility that fibrils formed in vivo may be structurally distinct. Efforts to prepare cryo-EM quality samples of brain-derived TDP-43 fibrils have not yet been successful.

While our long-term plans are to determine high-resolution structures of fibrils containing many different phosphorylation patterns, we hope the reviewer will appreciate that this is a major undertaking, especially since the efficiency of TDP-43 phosphorylation in vitro is low. The structure of fibrils generated from the non-phosphorylated protein provides a necessary foundation for future studies on structural impact of phosphorylation at individual Ser residues. We now provide a more detailed discussion of the issue of TDP-43 phosphorylation in the last paragraph of the Discussion section (p. 8/9).

With regard to the comment that only the structure of one fibril type (i.e., twisted fibrils) formed at pH 4 is solved, we wish to point out that morphologically these fibrils are indistinguishable from those formed at pH 6 (see our response to comment #1 of reviewer 1). The population of non-twisted fibrils is not amenable to structural characterization by cryo-EM.

2. It would be good to mentioned that it is an in vitro structure somewhere early in the main text.

As recommended by the reviewer, we now mention it in the Abstract and the last paragraph of the Introduction (p. 3)

3. How long is the full TDP-43 LCD? Only the length of the fibril core is mentioned.

The full-length TDP-43 LCD contains 148 residues, 139 of which are part of the amyloid core. We now provide this information in the second paragraph of the Discussion (p. 7).

4. Extended Data Fig. 5.: "Rrepresentative" one R to many; Also, it would be helpful to have the rest or at least the neighboring residues of full TDP-43 LCD (b,d) in light grey to help localize the fragment in the whole structure.

This is a very good suggestion. We have revised this figure (Extended Data Fig. 4 in the revised manuscript) as recommended by the reviewer. Also, the spelling error has been corrected.

REVIEWER COMMENTS

Reviewer #1 (Remarks to the Author):

The authors did a good job in addressing my concerns. I don't have further comment.

Reviewer #2 (Remarks to the Author):

I have no further suggestions for revisions. I support the publication of the manuscript.

Point-by point response to reviewers' comments

Reviewer #1:

The authors did a good job in addressing my concerns. I don't have further comment.

The reviewer requests no further revisions.

Reviewer #2:

I have no further suggestions for revisions. I support the publication of the manuscript.

The reviewer requests no further revisions.